# Evaluation of protein intake and protein quality in New Zealand vegans

**Bi Xue Patricia Soh**[1]*, **Matthieu Vignes**[1,2], **Nick W. Smith**[1], **Pamela R. von Hurst**[3], **Warren C. McNabb**[1]

**1** Sustainable Nutrition Initiative, Riddet Institute, Massey University, Palmerston North, New Zealand, **2** School of Mathematical and Computational Sciences, Massey University, Palmerston North, New Zealand, **3** School of Sport Exercise and Nutrition, College of Health, Massey University, Auckland, New Zealand

☉ These authors also contributed equally to this work.

* p.soh@massey.ac.nz

## Abstract

Dietary protein provides indispensable amino acids (IAAs) that the body cannot synthesise. Past assessments of total protein intake from vegan populations in western, developed countries were found to be low but not necessarily below daily requirements. However, plant-sourced proteins generally have lower quantities of digestible IAAs as compared to animal-sourced proteins. Simply accounting for protein intake without considering AA profile and digestibility could overestimate protein adequacy among vegans. This study quantified protein intake and quality, as compared to reference intake values among 193 NZ vegans using a four-day food diary. Protein and IAA composition of all foods were derived from New Zealand FoodFiles and the United States Department of Agriculture and adjusted for True Ileal Digestibility (TID). Mean protein intakes for males and females were 0.98 and 0.80 g/kg/day, respectively with 78.8% of males and 73.0% of females meeting the Estimated Average Requirement for protein. Plant-sourced proteins provided 52.9 mg of leucine and 35.7 mg of lysine per gram of protein and were below the reference scoring patterns (leucine: 59mg/g, lysine: 45mg/g). When adjusted to individual body weight, average IAA intakes were above daily requirements, but lysine just met requirements at 31.0 mg/kg of body weight/day (reference: 30mg/kg/day). Upon TID adjustment, the percentage of vegans meeting adequacy for protein and IAA decreased and only approximately 50% of the cohort could meet lysine and leucine requirements. Hence, lysine and leucine were the most limiting IAAs in the vegan cohort's diet. Legumes and pulses contributed most to overall protein and lysine intake. An increased proportion of legumes and pulses can potentially increase these intakes but must be considered in the context of the whole diet. AA composition and digestibility are important aspects of protein quality when assessing protein adequacy and is of particular importance in restrictive diets.

## Introduction

The plant-based (PB) diet transition has gained traction in recent years as one potential contributor to environmental sustainability and health. However, this discussion requires detailed

**Data availability statement:** All relevant data are within the paper and its Supporting Information files.

**Funding:** Lottery Health Project Grant (LHR-2022-185) PhD stipend from the Riddet Institute, Massey University.

**Competing interests:** The authors have declared that no competing interests exist.

**Abbreviations:** AA, Amino acid; AMDR, Acceptable Macronutrient Distribution Range; ASF, Animal-sourced food; BCAA, Branched-chain amino acid; BMI, Body mass index; BMR, Basal metabolic rate; BW, Body weight; EAR, Estimated Average Requirement; FAO, Food and Agricultural Organization of the United Nations; IAA, Indispensable Amino Acid; IAAO, Indicator amino acid oxidation; IQR, Interquartile ranges; MPB, Muscle Protein Breakdown; MPS, Muscle Protein Synthesis; NZ, New Zealand; PB, Plant-based; PSP, Plant-sourced proteins; RDA, Recommended Dietary Allowance; RDI, Recommended Dietary Intake; SD, Standard deviation; SAA, Sulphur-containing amino acid; TID, True-ileal digestibility; USDA, United States Department of Agriculture.

assessment of its nutritional quality, particularly for vegan diets which are the most restrictive PB diets. While most PB diets may include animal-sourced foods (ASFs), a vegan diet excludes all ASFs and their by-products, such as meat, dairy, eggs, and honey, and includes only foods with a plant origin [1–3]. As reviewed by Bakaloudi et al. (2021), globally, an increasing trend in the adoption of vegan diets was observed, especially in high-income countries and among the younger generation [1]. Recent analysis from the New Zealand (NZ) Health Survey however found that only 0.74% of a surveyed population (n = 23, 292) were self-declared vegans [4], a consistent finding with a previous survey on dietary behaviour among NZ adults in 2021 [5]. Notably, veganism is not a huge trend within NZ, but nutrient inadequacies due to unbalanced vegan diets lacking in fortification and supplementation requires attention [6,7]. Maintaining nutrient requirements and health in long-term vegan diets involve diverse food complementation - a practical approach to achieve adequacies in nutrients such as protein and amino acids (AA). While lower total protein intake has been recorded among vegans in some studies, consumption was not necessarily below daily requirements [1,8]. However, few studies evaluated AA intake among vegans and no studies have accounted for digestibility of AAs in the analysis [9]. Fundamentally, dietary sources of protein provide AAs, nine of which are classified as indispensable amino acids (IAAs) which cannot be supplied endogenously or have excess stored within the body over long periods [10]. Additionally, there is still no clear mechanism on whether the body can compensate imbalanced AAs with plasma pools [11]. IAAs are often available in lower or more varied concentrations in PB foods as compared to animal-sourced foods [12]. Furthermore, anti-nutritional elements in plant-sourced proteins (PSPs) may reduce the quantities of digestible (net absorbable) AAs [13,14]. Compared to diets that include ASFs, a diet consisting only of PSPs - especially if unbalanced, and lacking proper planning of complementary proteins – tends to supply lower concentrations of digestible AAs to efficiently meet individual requirements and are thus considered lower in protein quality [8,11,15].

Apart from protein source, quantity and amino acid composition, the anabolic response following protein ingestion is complicated by gastrointestinal amino acid absorption kinetics, dynamics in muscle perfusion and activation, variation across organs, individuality, as well as presence of physical activity [14]. Importantly, the body requires the supply of all IAAs to maximise muscle protein synthesis (MPS) and maintain an overall positive protein balance where muscle protein breakdown (MPB) is more inhibited than MPS. Consequently, within the context of a vegan diet, varied combinations of different PSPs consumed throughout the day are needed to improve IAA quantities in the plasma to threshold level to stimulate protein synthesis, grow and maintain muscle mass, as well as regulate other body functions [2,16,17]. Combining complementary protein sources to overcome limiting AAs in different protein sources can effectively improve the overall protein quality in the diet. Such combinations however require considerations in the trade-off between protein quality, energy, and fibre increment. Energy and fibre content are higher in many PB foods as compared to ASFs, with the latter contributing to satiety, placing limits on how much food a vegan can consume to meet nutritional requirements [18,19].

Taken together, protein quality (alongside other nutrient deficiencies) is a significant concern in vegan diets and must be analysed. Examining the current dietary status of long-term vegans is necessary to uncover potential limitations PB foods have in their supply of IAAs. Only with that can there be subsequent development of appropriate dietary interventions to support vegans in achieving nutritionally balanced diets with optimal protein quality. The main aim of this study was to determine protein adequacy, by quantifying overall protein intake as well as digestibility-adjusted amino acid profile of the vegan diet among a cohort of NZ vegans. These measurements are then compared to individual daily protein and IAA

requirements. A secondary aim was to explore how the dietary vegan pattern, in terms of food group contribution, impact protein intake and protein quality.

## Methods

### Study design

Dietary intake data was obtained from the Vegan Health Research Programme conducted at Massey University, Auckland, NZ from 2022 to 2023. The study was cross-sectional and aimed to explore vegan dietary patterns and determine the associations between nutrient intake and multiple health outcomes. Ethical approval was granted by the Health and Disability Ethics Committee (HDEC 2022 EXP 12312). All participants provided informed written consent prior to data collection.

### Study population

Using convenience sampling, healthy adults 18 years and above who live in NZ were recruited through university advertisements, local vegan societies as well as word-of-mouth. The inclusion criterion was that participants must have followed a vegan diet for at least two years. Pregnant and lactating women were excluded from the study. Recruitment began on 01/06/2022 and ended on 1/08/2023.

### Data collection

All participants visited the Human Nutrition Research Unit at Massey University on one occasion. Height and weight measurements were completed by trained researchers using the standardised operating procedures. Body height (cm) to the nearest 0.1 cm was measured without shoes with a stadiometer. Body weight (kg) to the nearest 0.1 kg was measured without shoes using floor scales. The Body Mass Index (BMI) kg/m$^2$ was calculated by taking the individual's body weight in kilograms divided by the square of his or her height in meters. Body fat percentage was assessed alongside bone tissue examination with dual energy X-ray absorptiometry (DXA).

Participants were instructed to provide a detailed record of all food, beverage and supplement consumption using a provided four-day food diary that spanned four consecutive days. Descriptions of records included the date and time of consumption, portion size, brand, variety, and cooking method. Completed records were returned to the research team for analysis.

### Data analysis

Food intake data was processed by supervised and trained Master of Dietetic students at Massey University using the FoodWorks Professional nutrient analysis software (Xyris, Australia, 2022) that is based on FOODfiles, the reference food composition table for NZ that provides values for various nutrients in NZ food items [20].

Protein was calculated as total nitrogen multiplied by a specific nitrogen-to-protein conversion factor [20]. As IAA composition data is not currently available in the food composition table for NZ, protein and IAA profiles for each food item was matched to the food composition data from the United States Department of Agriculture (USDA) [21]. When bulk recipes were recorded, the composite ingredients with their respective portion sizes were used to calculate total protein (g) and IAA (g) in the weighed recipe and then adjusted to the weight of the consumed portion. Within a home-made or commercial recipe, the ingredient that provided the highest source of protein - the food component of interest in this study - was selected and matched to the most similar food item in the USDA data.

To improve the quality in the estimations of nutrient intake and obtain the most appropriate match, food matching guidelines from FAO/INFOODS were followed [22]. In cases where an exact match was unachievable, the food item was matched to several possible food items to tabulate an averaged value for protein and IAA. To accommodate the influence of cooking methods on digestibility values for protein and AAs, a food item prepared in the most similar method from the USDA data was chosen as a match for the reported food item. To account for disparity between food compositions in the two countries, values of AAs from the USDA food composition data were normalised to the protein content of reported foods from NZ.

Values for protein and IAA were adjusted for digestibility using currently available TID values from the literature [23–25]. This involved categorising each food item into an appropriate food group that has an estimated digestibility coefficient (between 0 and 1) which is multiplied by the total protein and AA content of each food item [23–25]. To condense the number of food items for analysis, food items that contributed < 0.5 g of crude (unadjusted) protein in each of the four-day food diary were excluded. The remaining data captured more than 98% of the total cohort's protein intake and at least 92% of protein intake for every participant.

Percentage of energy from protein was calculated and expressed against the Acceptable Macronutrient Distribution Range (AMDR), which when using the reference body weight (BW) of 57 kg for women and 70 kg for men (using the mean estimated energy requirement of 36.5 kcal/day(d)) is between 10 to 35% of caloric intake or 1.05 to 3.67 g of protein/kg BW/d [26]. The use of AMDR incorporates protein intake as a portion of the daily energy [27,28]. The protein intake for each participant in g/kg of body weight was calculated by dividing their mean protein intake over four days by their BW. This was compared to the Australia and NZ Estimated Average Requirement (EAR) (0.68 g/kg BW/d for males and 0.60 g/kg BW/d for females) and to the Recommended Dietary Intake (RDI) (0.84 g/kg BW/d for males and 0.75 g/kg BW/d for females) for adults aged 19 to 70 years [28]. For older adults above 70 years, the EAR followed for males and females were 0.86 g/ kg BW/d and 0.75 g/ kg BW/d, respectively, and RDI for males and females were 1.07 g/ kg BW/d and 0.94 g/ kg BW/d, respectively [28].

To determine IAA contribution (mg) in one gram of protein, the quantity of each IAA (mg) consumed in a day was divided by total protein consumed in the day and the mean of four days derived for each individual. This allowed comparison of the IAA content of each food to its protein content (IAA mg/g protein). This IAA scoring pattern (mg/g protein) is a derivation of each IAA requirement (mg/kg BW/d) divided by protein or nitrogen requirement (g/kg BW/d) following the age group of 18 and above [29,30]. When incorporated with digestibility coefficients, protein quality of a specific diet could be examined [29].

To compare IAA intake to daily requirements (mg/kg BW/d), the quantity of each IAA (mg) consumed in a day was divided by each individual's body weight (kg) and compared to IAA requirement [29]. The percentage adequacy was calculated to determine proportion of participants meeting the daily EAR, RDI and IAA requirement. Measurements of body composition, BMI and percentage body fat were tabulated and explored alongside protein and IAA adequacy. Normal BMI range was established as 18.5 to 25 kg/m$^2$ for a NZ European population [31,32]. There is no clear definition of normal body fat percentage for males and females of different age groups for the NZ population, with only one study establishing normal body fat to be between 22% to 30%, although this is only for females [33]. The cut-off points of body fat in excess were more well-defined and references for overweight were taken to be 20.1 to 24.9% for males and 30.1 to 34.9% for females, and for obesity was equal or greater than 25% for men and 35% for females [34].

To determine how absolute IAA intake changes with total protein intake, linear regression of each TID-adjusted IAA (g) as a function of TID-adjusted total protein intake (g) was used. The value of the slope coefficient of the line was used to assess how IAA composition provided

by the vegan diet changes with overall protein intake. To evaluate how effectively individuals achieve daily IAA requirements relative to attaining protein requirements, the ratios of the actual intake values of each nutrient to their respective requirement was calculated. Ratios < 1 indicate that requirements were not met and ratios equal to or more than 1 indicate meeting and/or exceeding requirements. A heat map of four quadrants was used to visualise the distinct category of intake adequacy between protein and each IAA.

To assess the contribution of individual food groups to the percentage energy (MJ), dietary fibre (g), protein (g) and IAA (mg), each food item from the four-day food diary was matched to its respective food group, as defined in the 2008/2009 NZ Adult Nutrition Survey [35]. For example, wheat and cereal products were categorised as "Grains and pasta" while beans, peas and soy products were classified as "Legumes and pulses". Protein sources used in novel meat and dairy alternatives were matched to the food group of their principal protein-contributing ingredient [36]. To assess percentage contribution of foods to energy and dietary fibre, ingredients contributing most to energy and dietary fibre were identified and categorised into the relevant food groups.

All statistical analysis was conducted using R Studio (version 4.3.1). Continuous variables were described as mean and standard deviation (SD) and differences between sexes for energy and protein intake were calculated using the student's t-test. Differences were regarded as statistically significant when $p < 0.05$. Discrete variables were represented as counts with percentages.

## Results

### Study population

In total, 212 vegans participated in the study. Subjects who did not submit completed food records were excluded from the dietary analysis. A total of 193 participants consisting of 52 males and 141 females completed the food diary for at least 3 days and were included in the analysis. Table 1 describes the demographic and anthropometric characteristics of the study population. Females and New Zealanders of European descent form the majority. A large percentage of participants were highly educated with more than 80% completing tertiary education. Females had lower weight and height than males but were higher for percentage body fat.

### Energy and protein consumption between males and females

Table 2 presents the mean energy and TID - adjusted protein intake for males and females. Females had lower energy ($P < 0.01$) and protein (g/kg BW/ d) ($P < 0.05$) consumption compared to males. S1 Table in S1 File provides a summary of energy intake differentiated by sex and age group, compared with daily energy requirements. Percentage of caloric intake as protein is calculated as the AMDR. The results did not differ significantly between males and females ($P = 0.51$). When compared to the reference values for daily dietary protein intake, a larger proportion of males as compared to females met both the EAR and the RDI.

### Impact of TID adjustment on protein and IAA supply

Adjustments for TID showed clear differences for overall protein intake compared to requirements. Overall protein intake (g/kg BW/d) decreased from 1.25 g/kg to 0.98 g/kg for males and 1.05 g/kg to 0.80 g/kg for females, demonstrating that when TID is accounted for, mean protein is lower relative to reference values (Fig 1). After TID adjustment, the percentage of population meeting the EAR for protein adequacy decreased from 92.3% to 78.8% for males and 92.2% to 73.0% for females.

**Table 1. Characteristics of study subjects.**

| Characteristics | Male | Female |
|---|---|---|
| *n* (% of total cohort) | 52 (26.9) | 141 (73.1) |
| Ethnicity, *n* (%) | | |
| *European* | 45 (86.5) | 117 (83) |
| *Others* | 7 (13.5) | 24 (17) |
| Highest education level, *n* (%) | | |
| *Tertiary education* | 45 (86.5) | 129 (91.5) |
| *High school/ lower* | 7 (13.5) | 12 (8.5) |
| Duration of vegan diet in years, *n* (%) [1] | | |
| *2 to 4 years* | 84 (39.6) | |
| *5 to 10 years* | 93 (43.9) | |
| *More than 10 years* | 35 (16.5) | |
| Age in years | 41.3(11.9) | 39.5 (12.7) |
| Body weight in kg | 79.3 (12.5) | 66 (10.5) |
| Height in cm | 179.4 (7.9) | 166.6 (6.7) |
| BMI in kg/m² | 24.6 (3.0) | 23.8 (3.2) |
| Within BMI range, *n* (%) | 33 (63.5) | 92 (65.2) |
| Body fat in % | 22.9 (5.0) | 33.4 (6.0) |
| Overweight or obese for body fat, *n* (%) | 35 (67.3) | 100 (70.9) |

[1]Based on consolidated data of 212 subjects. Continuous values are expressed as mean (SD) unless otherwise stated, categorical values are expressed as counts, *n* (percentage). BMI normal range is established at 18.5 to 25 kg/m². Cut-off points of body fat for overweight was taken to be 20.1 to 24.9% for males and 30.1 to 34.9% for females, and for obesity was equal or greater than 25% for men and 35% for females [34].

**Table 2. Energy and TID-adjusted protein consumption among vegan participants.**

| Intake | Male | Female |
|---|---|---|
| Energy intake in MJ/d | 10.4 (2.6) | 7.5 (1.9) |
| Protein intake in g/d | 76.3 (27.5) | 52.1 (16.3) |
| Protein intake in g/kg BW/d | 0.98 (0.36) | 0.80 (0.27) |
| AMDR for protein in % | 12.4 (3.5) | 12.0 (3.7) |
| **Percentage meeting (%)** | | |
| EAR | 78.8 | 73.0 |
| RDI | 59.6 | 53.9 |
| AMDR for protein | 82.7 | 72.3 |

Intake values were expressed as mean (SD). EAR and RDI for males and females were obtained from the Nutrient Reference Values of Australia and NZ [28]. AMDR for protein reference value was obtained from literature [26].

Table 3 provides a summary of IAA consumption compared to the AA scoring patterns (mg/g of protein) and daily AA requirements (mg/kg BW/d). When compared to the AA scoring pattern (mg/g), the analysed diets would supply lower lysine, leucine, isoleucine and valine per gram of protein. However, when adjusted to individual body weight, all IAAs met AA requirements in mg/kg BW/d.

TID adjustment decreased all IAA intakes (mg/kg BW/d) (Table 3). Before adjustment, the percentage of individuals meeting adequacy for histidine, threonine, and tryptophan was close to 100%, while percentage adequacy for leucine and lysine was lower. Digestibility adjustments

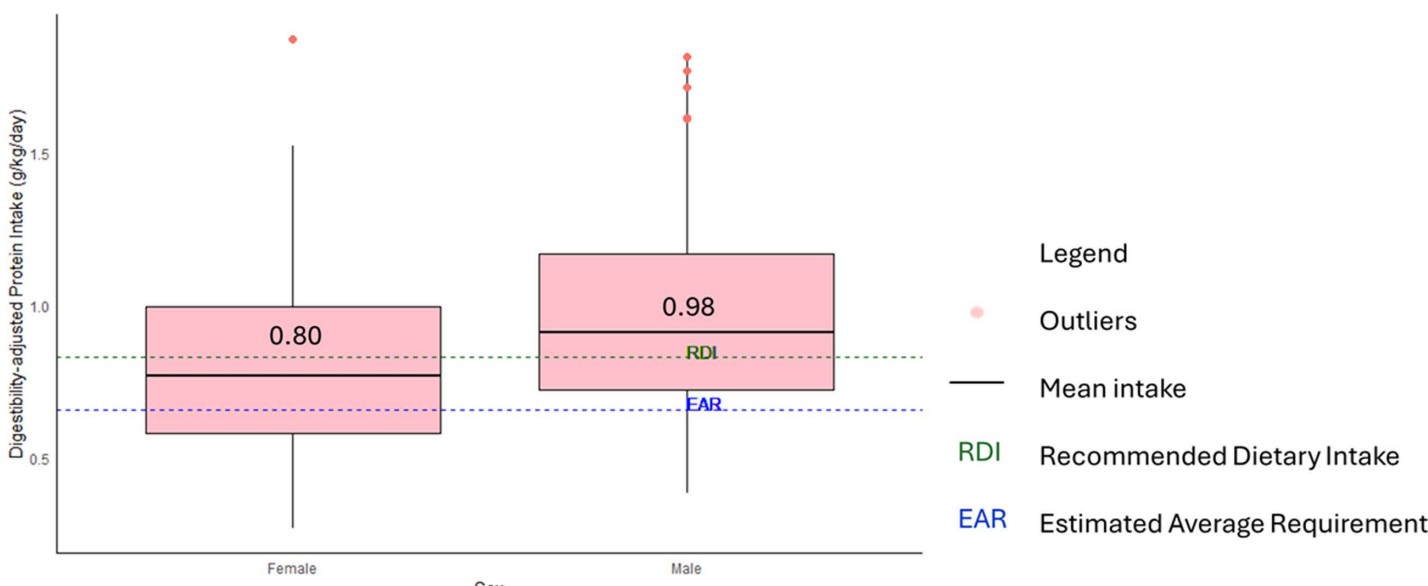

**Fig 1. TID-adjusted mean protein intake for males and females.** The mean is represented by the horizontal black line and numeric value in each box. The box represents the IQR and the whiskers show the 10th to 90th percentile range. Outliers in the study consumed protein above 1.70 g/kg/d.

**Table 3. AA consumption patterns as compared to AA scoring pattern and reference AA requirements, before and after TID adjustments.**

| | His | Ile | Leu | Lys | SAA[4] | AAA[5] | Thr | Trp | Valine |
|---|---|---|---|---|---|---|---|---|---|
| **AA scoring pattern (mg/g protein requirement)** | 15 | 30 | 59 | 45 | 22 | 38 | 23 | 6 | 39 |
| **Reference AA requirements (mg/kg/d)** | 10 | 20 | 39 | 30 | 15 | 25 | 15 | 4 | 26 |
| **Unadjusted values** | | | | | | | | | |
| *AA intake (mg/g protein)* | 18.2(2.8) | 28.4(4.3) | 50.8(7.3) | 35.2(5.9) | 22.5(3.5) | 55.6(7.9) | 25.7(3.8) | 8.4(1.7) | 34.5(5.3) |
| *AA intake (mg/kg/d)* | 20.1(7.7) | 31.2(11.6) | 55.9(20.8) | 39.3(16.0) | 24.8(9.3) | 61.1(22.9) | 28.3(10.5) | 9.1(3.5) | 37.8(13.8) |
| *Percentage adequacy (%)[1]* | 96.4 | 84.0 | 75.6 | 66.3 | 88.6 | 99.5 | 95.3 | 97.9 | 78.2 |
| **Digestibility-adjusted values [2]** | | | | | | | | | |
| *AA intake (mg/g protein [3])* | 19.1(2.8) | – | 52.9(7.6) | 35.7(6.2) | 24.0(3.8) | – | 26.1(3.9) | 8.5(1.8) | – |
| *AA intake (mg/kg/d)* | 16.4 (6.5) | – | 45.0(17.5) | 31.0(13.6) | 20.3(7.6) | – | 22.1 (8.5) | 7.0(2.7) | – |
| *Percentage adequacy (%)* | 86.5 | – | 56.0 | 43.5 | 71.5 | – | 78.2 | 90.0 | – |

His, histidine, Ile, isoleucine, Leu, leucine, Lys, lysine, SAA, sulphur amino acids, AAA, aromatic amino acids, Thr, threonine, Trp, tryptophan, Val, valine.

AA scoring pattern mg/g protein requirement and reference AA requirements (mg/kg/d) are obtained from FAO [30]. Values of AA intake (mg/g protein) and AA requirement (mg/kg/d) are expressed in mean (SD). Dashes are indicated for Ile, AAA, and valine in TID-adjusted values, because digestibility values of food for these AAs are not currently available.

[1]Percentage adequacy describes the percentage of the study population consuming each AA at or above the AA requirements (mg/kg/d).

[2]Digestibility coefficients were available for histidine, cystine, methionine, leucine, lysine, threonine and tryptophan [23–25].

[3]Total protein (g) was multiplied by digestibility coefficient [23–25].

[4]SAA, sulphur-containing amino acids, which are cystine and methionine. Intake values for cystine and methionine were added at each day for each participant.

[5]AAA, aromatic amino acids, which are phenylalanine and tyrosine.

decreased the percentage adequacy for all IAAs. Notably, the decrease for lysine and leucine was by at least 15%. Consequently, only approximately half of the cohort met IAA requirements for lysine and leucine. The impact on utilisable AA quantity following TID adjustment is also observed in Fig 2, as the mean value of each AA (mg) reduced after digestibility adjustment.

## Comparison of TID-adjusted IAA intake to reference patterns and daily requirements

Table 3 and Fig 3 show that while the other TID-adjusted IAAs were meeting reference scoring patterns, TID-adjusted lysine and leucine were below their reference scoring patterns of 59 mg/g of protein and 45 mg/g of protein, respectively. This indicates that lysine and leucine are the two most limiting AAs as provided by PB protein sources in the vegan cohort.

Mean IAA intake adjusted to BW met daily IAA requirements, but for lysine, the intake (31.0 mg/kg/d) only just met requirement (30 mg/kg/d) (Fig 4 and Table 3). Approximately half the cohort did not meet BW-adjusted requirements for digestible lysine (56%) and leucine (43.5%) (Fig 4). In investigating protein intake of individuals that did not meet both lysine and leucine requirements, 56.6% did not meet the protein EAR (g/kg BW/d).

BMI and percentage body fat were explored to assess the relationship between body composition and protein particularly for the two most limiting IAAs, lysine, and leucine. The results are presented in S2 and S3 Tables in S1 File. Approximately 50% of the group inadequate in TID-adjusted leucine (g/kg BW/d) had BMI above the normal BMI range. Approximately 80% were in the overweight or obese range for body fat. Among the individuals inadequate for TID-adjusted lysine, 42% had a BMI above the normal range and 85% were in the overweight and obese category for body fat. The energy intake and AMDR for protein

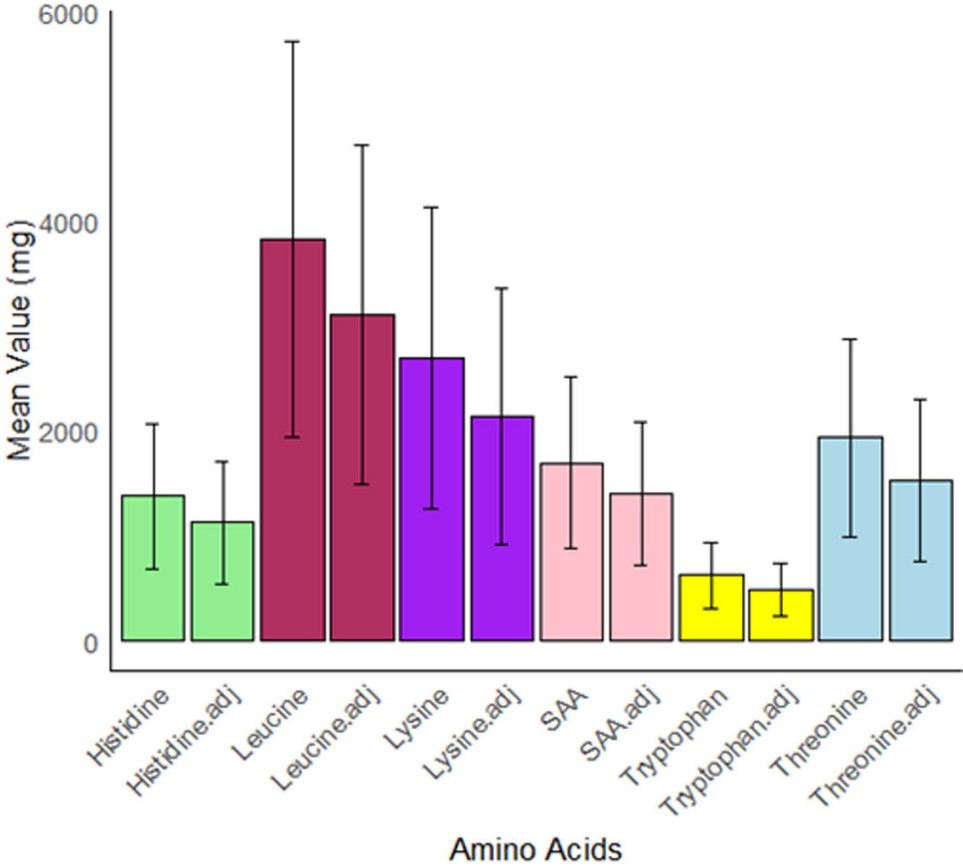

**Fig 2. Comparison of mean daily IAA intake quantity before and after digestibility adjustments.** Error bars represent standard deviations. adj, AA TID-adjusted values; SAA, sulphur-containing amino acid.

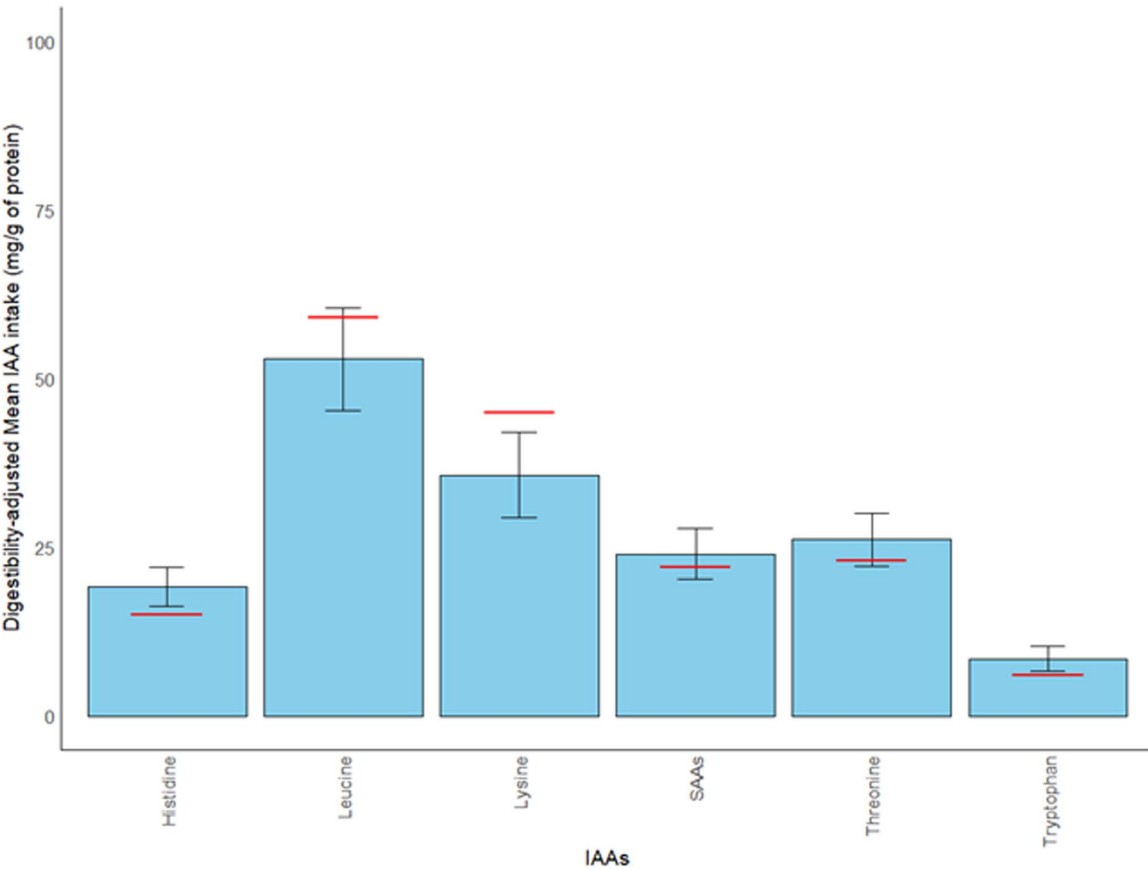

**Fig 3. Comparison of each IAA intake with its respective AA scoring pattern.** Scoring pattern is represented by the red horizontal lines. The error bar represents the SD. Blue bars that meet or are above the red line indicate that mean IAA intake pattern in mg/g of protein met the AA scoring pattern.

were also examined in individuals that were below the requirements for leucine and lysine (S3 Table in S1 File). Mean energy intake and mean AMDR for protein were lower in this group as compared to the cohort.

Only a small percentage (7.2%) of participants were below requirement for all the digestibility-adjusted IAAs. BMI was above the normal range for all but two females, and the percentage body fat was in the overweight and obese category for all individuals.

Regression models were applied to determine how increasing the quantity of TID-adjusted protein intake is associated with the change in the quantity of each TID-adjusted AA (Fig 5). Generally, an increase in each IAA intake is associated with an increase in total protein intake. Wider variation in the range of IAA intake is observed with total protein intake above 50 g. The coefficient of determination as denoted by the $R^2$ value is similar for all IAAs and total protein, except for tryptophan which has the smallest $R^2$ value of 0.64, showing that protein and tryptophan intakes have poorer association as compared to that of protein and the other IAAs. Greater variability in tryptophan levels is observed at elevated protein levels as compared to other IAAs.

Fig 6 demonstrated how participants were distributed across each distinct category of intake adequacy for protein and each respective IAA. More vegans could easily achieve the requirements for IAAs like histidine, tryptophan, threonine, and SAAs, but less easily

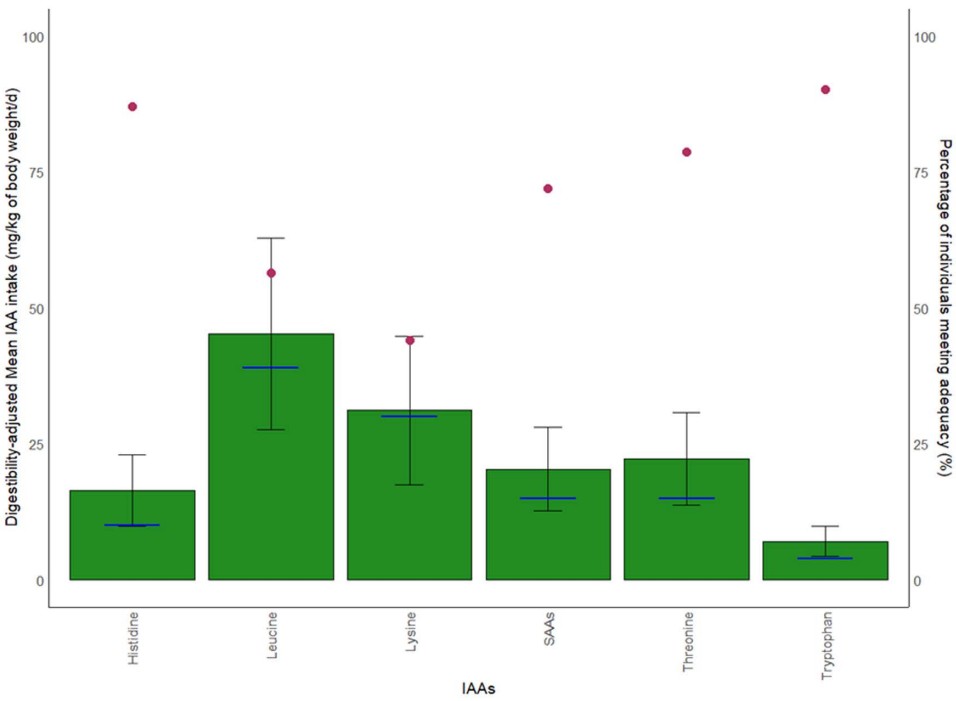

**Fig 4. Comparison of TID-adjusted IAA intake compared to its respective requirement.** The requirement is represented by the blue horizontal lines. Each green bar represents the mean IAA intake in mg/kg BW/d (left-hand vertical axis) and the error bar represents the SD. Red dots represent percentage of individuals in the cohort meeting daily IAA adequacy (right-hand vertical axis).

achieve the requirements for lysine and leucine. For example, when protein intake is above adequacy, more individuals were above histidine adequacy (n = 142) as compared to lysine (n = 84) and leucine (n = 108). The largest numbers below adequacy even at higher protein consumption (above adequacy) were found in the quadrants for lysine (n = 61) and leucine (n = 37).

## Contribution of food groups to nutrient profile

Fig 7 shows the percentage contribution of each food group to the intake of energy, dietary fibre, TID-adjusted protein, and TID-adjusted IAA. The highest contributing food group to total energy intake was "grains and pasta" (37.3%), followed by "legumes and pulses" (26.8%) and "nuts and seeds" (14.2%). The highest contribution to total protein intake and total lysine intake was provided from "legumes and pulses" (43.9% and 56.4%, respectively). In contrast, "grains and pasta" provided lower contributions to total protein intake (30.4%) and total lysine (21.0%).

Analysis of consumption patterns of novel PB foods as compared to traditional sources was out of the scope of this study but protein and IAA contribution from novel foods was estimated. These foods were assumed to serve as meat and dairy replacements in the diet of this vegan cohort. Ingredients used in novel PB alternatives varied but extruded protein sources from soy and pea (e.g., isolates) were the main protein-contributing ingredient from PB meat alternatives and powdered supplements. The main contributing food group to the novel alternatives was "legumes and pulses" which supplied more than 90% of total TID-adjusted protein and lysine from this group.

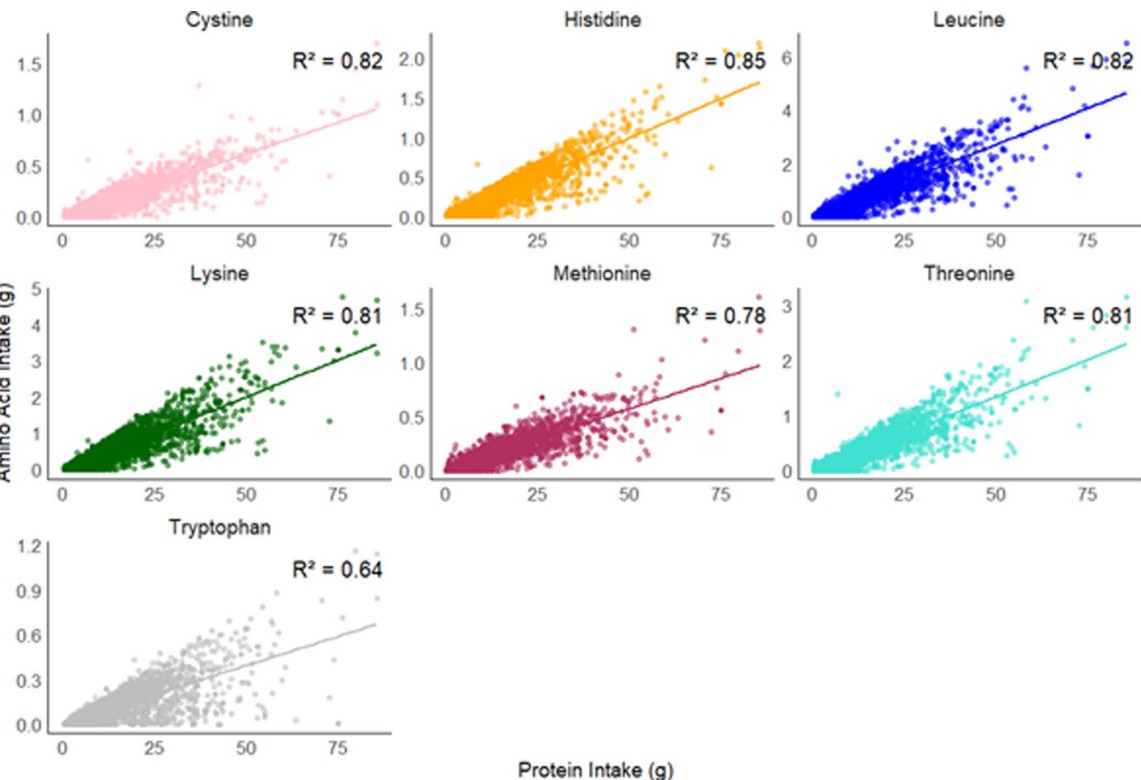

**Fig 5. Regression analysis describing relationship between TID-adjusted total protein (g) and each TID-adjusted IAA consumption (g).** Each IAA is represented by a different colour. Data points represent all individuals in the cohort, for each day of the food diary. The $R^2$ value quantifies the proportion of variance in IAA intake with increasing protein intake. Higher $R^2$ value indicates stronger relationship between the two variables. $R^2$ values: protein and cystine: 0.82; protein and histidine: 0.85, protein and leucine: 0.82, protein and lysine: 0.81; protein and methionine: 0.78; protein and threonine: 0.81; protein and tryptophan: 0.64.

## Discussion

The present study assessed the digestible protein and IAA intake in a cohort of vegan individuals in NZ. To our knowledge, this is the first comprehensive dietary analysis for protein adequacy in long-term vegans residing in NZ. This study adds to past findings of protein and IAA intake assessments among vegans from other regions of the world [37–41] and contributes the perspective of protein quality assessment relative to daily requirements. The key finding in this study was that lysine followed by leucine were the two most limiting IAAs in the vegan diet of this NZ cohort, with a large proportion of the cohort consuming insufficient amounts despite most individuals achieving adequate daily protein intake. Assessments for protein quality must consider protein digestibility, which is often lower from PB foods. As evident from this study, digestibility adjusted calculations decreased protein and IAA values compared to using a composition-only approach.

### TID-adjust Protein and IAA intake relative to requirements

As observed in Fig 1, total protein intake is of small concern within the vegan cohort as, even after digestibility adjustments, mean protein intake for both males and females met the EAR. However, only 55.6% of individuals who met this protein requirement achieved adequacy for all digestibility-adjusted IAAs, showing that protein quality is not always assured even when total protein is adequate. Moreover, the EAR and RDI values are derived from nitrogen

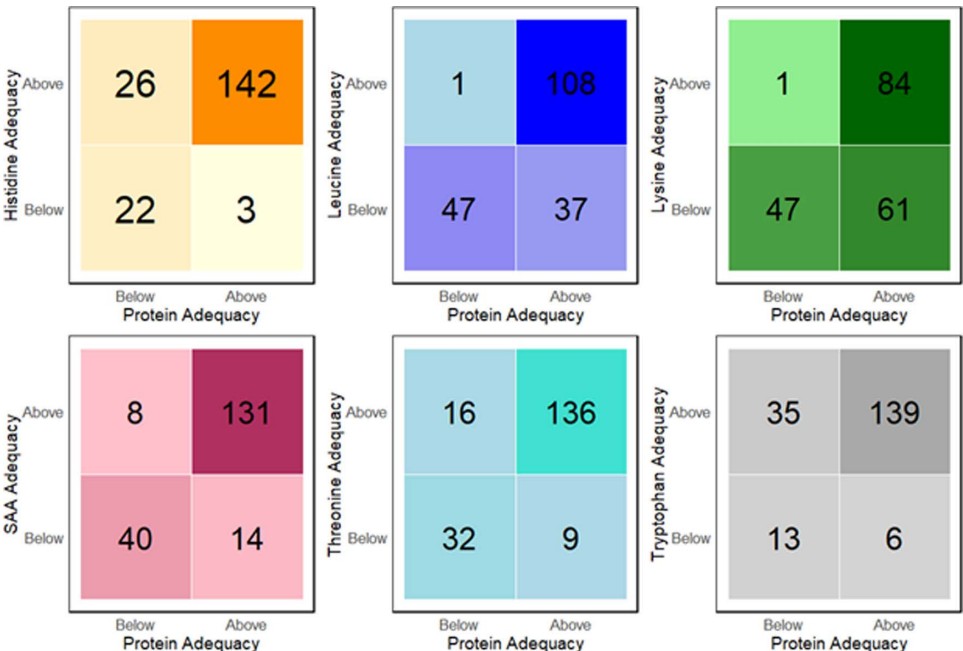

**Fig 6. Visualisation of quadrants for the comparison of each IAA adequacy relative to protein adequacy.** Each colour gradient is representative of an IAA (histidine, leucine, lysine, SAA, threonine, and tryptophan) which have all been adjusted for digestibility. The numbers on the quadrants represent the number of vegans distributed for each distinct combination of IAA and protein adequacy. For example, in histidine quadrants, 142 individuals achieved adequacy for both total protein and histidine, when compared to daily requirements adjusted to BW. Adequacy is expressed as a ratio of intake/ requirement. Ratio < 1 indicates below requirement and ratio equals or more than 1 indicates meeting and above requirement. The total number of vegans represented in each quadrant collection is 193.

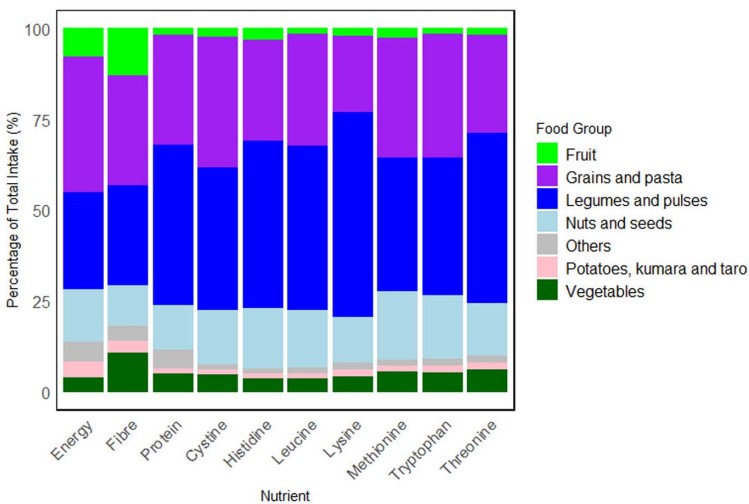

**Fig 7. Contribution of food groups to energy, dietary fibre, TID-adjusted protein, and TID-adjusted IAA intake.** "Others" indicate food groups such as "Fats and oil", "Sugar and sweets", "Savoury sauces and condiments".

balance studies based on the attainment of the lowest possible AA oxidations or minimal value for nitrogen balance, with excess AA intake unnecessary for metabolic advantage [17,42]. More recent work with indicator amino acid oxidation (IAAO) methods, although only conducted on young males, noted an increase of the Recommend Dietary Allowance (RDA) of 0.8 g/kg/d of protein to approximately 1.2 g/kg/d to cater to functional properties of the AAs [2,17,26,43]. These conclusions were still based on high-quality protein that can provide adequate IAA to match requirements [2,44]. A diet consisting entirely of PB foods may require an elevation of the recommended daily protein requirement to meet metabolic needs [45]. If so, the vegans in this present study, especially the females with a mean protein intake of 0.80 g/kg/d, will be below RDA requirements. A different protein requirement specified for vegan populations is currently absent but should be considered due to its restrictive nature. Considering protein quality may be superfluous in diets that provide protein sources of high biological value. However, in diets that exclude all ASFs, simply comparing to EAR or RDA values and assuming adequacy will neglect the overall digestible AA profile and its impact on adequacy [46].

Protein quality assessments among vegan populations, even without adjustments for digestibility, have also found reduced intakes of total lysine and branched-chain amino acids (BCAAs) such as leucine, when compared to omnivorous diets [41,47–51]. A similar cross-sectional study on 40 Danish vegans also found that lysine was the most limiting IAA - only 50% of participants met daily requirements – followed by SAAs and leucine as the next limiting IAAs [52]. While the SAAs appear to be adequate in our results on the present NZ vegan cohort, the percentage of individuals matching lysine and leucine requirements (mg/kg BW/d) were the lowest (Table 3 and Fig 4). As these AAs have important metabolic functions, such as DNA methylation, bone health and particularly for leucine, MPS [53–55], deficiency will have negative implications for long-term health.

When adjusted for individual body weight, intake for all IAAs adequately met daily requirements, even for lysine and leucine, albeit at lower levels than the other IAAs. Currently, body weight is used as the predictor for, and assumed to have a linear relationship with human protein requirement [56,57].

## Exploration of body composition and its impact on protein and IAA intake

The group which was adequate for protein intake when compared with EAR (g/kg BW/d) had higher energy intake as compared to the group that was inadequate for protein (S2 Table in S1 File). This suggests higher energy consumption per day may contribute to higher protein consumption, as similarly depicted by higher AMDR values (S2 Table in S1 File). Interestingly, the group that was inadequate for protein had higher weight, BMI, and body fat despite consuming lower energy intake than the group adequate for protein.

Individuals in this study, who met IAA requirements, were of normal BMI. This could indicate a healthy or lower body weight. Based on the EAR of 0.66 g per kg of body weight, individuals with lower body weight would meet body weight requirement of protein at a lower quantity. Concurrently, sufficient high-quality protein intake in these individuals who met IAA requirements could have a role in promoting satiety, maintaining healthy weight and lean muscle mass [11,58]. A study with NHANES data on American adults similarly found an inverse correlation with BMI and protein intake and attributed this finding to mechanisms in lipid and glycaemic regulations [59]. Another study also found 7% lower BMI in individuals consuming higher energy and higher protein intake [11].

In individuals, who were below leucine and lysine requirements but had higher BMI and percentage body fat (S3 Table in S1 File), underreporting of energy intake could be a problem. Yet, our results could also reflect true intake deficiency of overall food intake and consequent

reduced dietary protein intake (57.1% under EAR), the latter being key for lean muscle (constituted by fat free mass) protein synthesis [60,61]. Since adipose tissue contributes less to basal metabolic rate (BMR) than lean body tissue, these individuals may have lower energy requirements [62] and possibly lower metabolic rate. However, there is insufficient evidence from the data in this research to validate this. While the relationship between body composition and protein requirement or intake is an important area of research, it is not the main focus of this study.

In developed countries, protein requirements could generally be achieved with adequate energy intake and diverse diets [63]. This is not always the case in diets, whose protein density to energy is low, such as in an unbalanced vegan diet. Lower AMDR for protein (~10%) among individuals below lysine and leucine requirements (S3 Table in S1 File) in this cohort may suggest unbalanced macronutrient distribution, which could attribute to poorer body composition outcome (higher BMI and percentage of body fat) but further analysis of food group contribution to other carbohydrates and fats is required.

Clinically significant differences in protein requirement tested with different approaches, such as body weight and fat free mass, exist but neither method has been proven to be more accurate [64]. Geisler et al. also described that body composition, which has larger variability, lead to large variances in protein intake recommendations as compared to using only body weight [57]. There are currently no cut-off values of protein intake recommendations based on body composition metrics, like fat free mass. Clearly, more large-scale and well-controlled studies with comparisons between individuals of different long-term diets could provide more precise guidance on protein and IAA intake based on body requirements and composition, rather than solely relying on body weight. Ascertaining the effect of varied protein sources and macronutrient distribution on body composition would be valuable but is not the emphasis of this study.

## Food group contribution to protein and IAA

Fig 7 demonstrated that "legumes and pulses" was the food group providing the largest part of overall protein and lysine intake. This was similarly reported in a systematic review of four plant-based diets which found legumes, including soy contributing most to total protein intake in vegans as compared to other dietary patterns [65]. Plant sources higher in leucine content include dried seaweed, roasted pumpkin seeds, dry-roasted soybeans, cooked lentils and dry-roasted peanuts [66]. These foods may be consumed in insufficient quantities in the cohort or in forms that limit digestibility. To improve total protein and lysine intake to match requirements, the intake of legumes and pulses could be increased, which could increase the protein to energy ratio of the diet. This approach must however consider the implications on dietary fibre and its contribution to satiety, as well as the overall balance of other nutrients.

Cereals such as wheat and maize contribute less to digestible lysine. Moreover, as indicated by Fig 7, SAAs are more limiting in legumes like soy, peas, beans and lentils [67,68]. Hence, the strategy of complementing legumes and cereals within each vegan meal to supply more balanced profiles for SAAs and lysine could be an effective approach for this vegan cohort. This is however not always straightforward, as these AAs vary substantially across many PB foods as compared to ASFs. For example, the methionine:cysteine ratio varies across maize and wheat even if they are categorised in the same food group [69]. Furthermore, one analysis suggested that pairings of PB foods should be conducted on an individual food level rather than broader food categories, which may offer low specificity in predicting IAA patterns [68]. This implies that studies working on modelling nutritionally balanced vegan diets must critically examine the different permutations of PB foods in the appropriate proportions.

The current dietary pattern of this vegan cohort compared poorly to the IAA scoring patterns for lysine and leucine, which indicates a poorer match between IAA intakes from the current vegan protein sources and the body's needs [29,70,71]. Again, these AA profiles were based on the minimal or safe protein (g/kg BW/d) for nitrogen balance [70]. It is thus unclear if these values should be increased in diets of low protein quality.

## Optimising total protein and IAA intake

Larger variation as observed at higher protein intake (>50g) in Fig 5 could be attributed to fewer counts of individuals who consume protein intake above this level but generally, increased total protein consumption is associated with increased absolute IAA intake consumption. Absolute tryptophan intake appeared to have the weakest association with protein intake ($R^2$ = 0.64). This could be related to lower content of tryptophan in protein structures, such as in cereals and maize [72,73]. In corroboration, studies have found decreased intake of tryptophan by 50% in both male and female individuals on vegan diets as compared to an omnivorous diet [41,47]. Another reason could be related to methods used in tryptophan analysis, and these are addressed in the limitations.

Higher numbers in the top right quadrant for all IAA and protein adequacy comparisons in Fig 6 indicated better achievement of digestible IAA requirement for some IAAs, when protein requirement was meeting or above requirements butlysine and leucine adequacy was more difficult to achieve as compared to the other IAAs. This was not unexpected as these IAAs, particularly lysine, have low concentration in cereals as compared to other PB foods such as legumes [74]. To follow up from this study, the examination of the most appropriate PB protein combination and proportion in individuals across the quadrants is crucial to optimise both protein and IAA intake to achieve daily requirements. Investigating dietary patterns among individuals at risks of deficiencies (below requirements) is another critical step.

Overall, we have shown that, while the majority of vegans in the cohort were meeting total protein intake compared to the EAR, about a quarter had insufficient intake (Table 2), as similarly reported in past surveys on vegan populations [8,41]. Beyond examining total protein, the critical examination of individual AAs is of greater importance due to their direct roles in protein metabolism and other chemical pathways within the body. Uncovering potential deficiencies in lysine and leucine within this vegan cohort is the key finding of this study. Simply increasing protein intake alone is insufficient to ensure adequacy of all IAAs and may result in unrealistic food weight or calories consumed. Diverse PSPs are required to compensate limiting IAAs in different plant foods and allow simultaneous delivery of IAAs required for metabolic function [11].

Generally, the vegans in this study followed diets inclusive of varied plant foods, which could imply moderate acceptability and knowledge of diverse food combinations. This could be related to a high level of education among participants, but also requires further confirmation with nutritional knowledge scores. Yet, IAA intake profiles (especially for lysine and leucine) could reflect inappropriate proportions of cereals to legumes. Potentially, exploring the optimisation of cereal to legume ratio can improve leucine and lysine profiles. Food combinations occur at the meal level so these strategies must be implemented within such contexts. These methods must however account for overall macronutrient and micronutrient balance, and consider the impacts of dietary fibre intake, which contributes to fullness and gastro-intestinal discomfort in excess [75].

Novel PB meat and dairy alternatives of leguminous sources could offer opportunities to narrow the dietary gaps in protein and certain micronutrients, as seen by the lysine contribution from these foods in this study. However, novel alternatives have highly variable ingredient and nutrient composition, as well as processing methods, which have differing impacts on

digestibility [76]. A review of product formulations is necessary to uncover how limiting IAAs and overall nutritional content may be improved in novel alternatives.

## Strengths and limitations

High protein quality diets cannot be assumed in vegan populations even if total protein intake meets daily requirements. Varied digestibility among plant-sourced proteins must be accounted for, and this is the first strength of this study. Digestibility scores provided from the literature may however be imprecise due to simplified classifications of many food items into one food group. Neither have all food preparation techniques been differentiated for digestibility variations. Methodological inconsistencies in obtaining digestibility values for tryptophan could further explain the wider variance ($R^2$ = 0.64) observed at higher protein intake (Fig 5). Firstly, tryptophan cannot be analysed by standard amino acid hydrolysis which degrades its labile indole ring and even with alkali hydrolysis, only small quantities (16 to 20%) can be recovered [77–80]. Secondly, varying protein sources could explain the variation in digestibility values of tryptophan [81]. Finally, tryptophan digestibility is impacted by competition from large neutral amino acids and terminal ileum absorption kinetics which also vary across protein sources – higher tryptophan absorption is strongly correlated with higher tryptophan concentrations in foods [82].

The pig is the most valid animal model to obtain TID values for a large range of protein sources but digestibility data from many fruits and vegetables are still lacking [81]. However, this may be of lower consequence in our study as these foods do not contribute high quantities of protein and IAAs. Effort has also been made in this study to distinguish several common PB foods. For example, instead of simply categorising novel PB meat alternatives using extruded pea and soy protein into 'peas' and 'soyabeans', they have been adjusted for digestibility using more recently evaluated digestibility calculations from pea and soy concentrates and from isolates [24]. While this step does not account for all aspects of bioavailability such as the overall food matrix and intra-individual absorption and utilisation variations, it indicates the proportion of nutrient that is digestible [30,83]. This also accounts for differences between traditional and more novel plant foods made with protein isolates and concentrates, with the expectation that more specific measurements will be used when updated databases become available.

Misreporting is always an inherent limitation in dietary records with studies finding underreporting of carbohydrates, protein, and snack foods among adults [84], which could account for low energy intake in this cohort. Little information is however known about which foods are misreported within each macronutrient group [84]. A similar problem was highlighted by Waldman et al. (2003) who found a large proportion of surveyed German vegans falling below the recommended daily energy requirement, even when lower values were used as reference [85]. Without a controlled feeding study to assess the accuracy of self-reported recalls, it is unclear which foods may be misreported in this study. However, the use of the four-day food diary on long-term vegans is a strength in this study as the calculation of the mean protein and energy intake over four days allows for correction of intra-individual variations. For outliers in this study, the food diaries were checked for accuracy and anthropometric measurements were validated. Exclusion of outliers did not impact significance of results. For the context of checking if these outliers achieved a high protein quality vegan diet alongside a high protein diet, they were not excluded from the analysis. Another limitation was at the calculation step of protein content in food, where the nitrogen-to-protein conversion factor of 6.25 was used, based on the assumption that all proteins contain 16% nitrogen [86]. This is imprecise but appears to be the current practice when quantifying protein.

We also acknowledge an over-representation of females in this study, who have lower energy and total protein intake than males in our results and could potentially bias overall

energy and protein intake of the cohort. However, the sampled population reflects the typical demographics of the NZ vegan population where the ratio of females to males is high and represents the vegan population in NZ which constitutes less than 1% of a surveyed population [4]. Similarly, few studies on vegan diets evaluated populations of vegans above 100 participants and were mostly dominated by females [1]. Nevertheless, longer-term studies with more balanced ratios of males and females, and larger cohort sizes of long-term vegan populations are necessary to validate these results. Limitations were also present with body composition measurements which could be improved with repeated DXA measurements, but this must account for participant burden. While most individuals in this cohort were of European descent and the BMI for the general NZ population is provided, the standard of measurement may differ for that of other ethnic groups [33]. Lastly, we have not evaluated plasma AA levels as compared to other studies which have compared intake and metabolite measurements [41,50,87,88]. This was out of the scope of this study but is a potential area of exploration to ascertain if insufficient intake of IAAs would manifest in nutritional status.

## Conclusion

Generally, higher consumption of total protein to achieve daily protein requirement is associated with higher protein quality and achievement of daily IAA requirements. However, these conclusions vary across different IAAs and could be impacted by the combination of different plant protein sources. Meeting adequate total daily protein intake in a vegan diet does not always guarantee a high protein quality diet and simply considering total protein intake without delving into protein quality will overestimate protein adequacy among vegans. Digestibility-adjusted leucine and lysine were the two most limiting AAs when compared to AA scoring patterns and daily requirements in this cohort of NZ vegans. Given their critical metabolic functions, dietary recommendations to improve the overall IAA composition and increase leucine and lysine intake need to be considered. Potentially, increasing intake from legumes and pulses could improve the overall IAA profile but determination of ratios between cereals and legumes needs to be carefully examined for energy, dietary fibre and overall nutritional profile.

This study showed the importance of accounting for digestibility in dietary studies, as simply utilising IAA composition of food items can overestimate protein adequacy. As the body is incapable of producing or storing excess IAAs, there are limited time windows throughout the day for effective protein utilisation. A few studies have discussed the potential of an even distribution of protein and AAs in meals to maximise muscle protein accretion [17,61,89]. To this end, assessing protein intake and protein quality of different plant food combinations would be more suitable at the meal level. However, there is still no consensus on the appropriate protein and IAA requirement for each meal, and whether this may differ among sexes, age groups and body composition patterns. To improve protein quality in vegan diets, future research should focus on identifying optimal combinations of traditional and novel PB foods to achieve balanced intakes of all digestible IAAs, particularly for lysine and leucine, and this should ideally be examined within the context of meals. Although the global vegan population remains relatively small, it is important to generate evidence-based knowledge to ensure such diets have optimal quality due to the restrictive nature of consuming only plant-sourced foods.

## Supporting information

**S1 File.  S1 Table:** Energy intake as distinguished by sex and age.[1] PAL, Physical activity level. Energy requirement values are obtained from Nutrient Reference Values of Australia and New Zealand which provided low to moderate activity range (28) and based on 76 kg male and 61 kg female. Percentage adequacy is represented as individuals who are at or above the low

range of energy requirements. For males, this would be 10.8 MJ (19 - 30 years), 11 MJ (31 - 50 years), 9.5 MJ (51 – 70 years), 7.4 MJ (above 70 years), and for females, this would be 8.1 MJ (19 - 30 years), 7.9 MJ (31 - 50 years), 7.6 MJ (51 – 70 years) and 7.1 MJ (above 70 years). Dash lines represent not applicable calculations as there are no males above 70 years of age in the study. **S2 Table:** Protein intake and body composition. n represents number of individuals. Values represented as mean (SD) unless stated. Adequacy for protein is derived by comparing individual protein intake in g/kg BW/d with the EAR provided by the Nutrient Reference Values of Australia and New Zealand (28) which is 0.60 g/kg/d for females and 0.68 g/kg/d for males, between 19 and 70 years of age, and 0.75 g/kg/d for older females. **S3 Table:** Energy and AMDR of individuals who were below daily requirements for lysine and leucine. *n* represents number of individuals. Energy requirement values are obtained from Nutrient Reference Values of Australia and New Zealand (28) and based on 76 kg male and 61 kg female. Percentage adequacy is represented as individuals who meet or are above the low range of energy requirements. For males, this would be 10.8 MJ (19 - 30 years), 11 MJ (31 - 50 years) and 9.5 MJ (51 – 70 years). For females, this would 8.1 MJ (19 - 30 years), 7.9 MJ (31 - 50 years) and 7.6 MJ (51 – 70 years). Dash lines represent not applicable calculations as there are no males above 70 years of age, and no females above 70 years of age that are inadequate for TID-adjusted lysine and leucine in the study. AMDR, Acceptable Macronutrient Distribution Range is for protein, which is between 10 to 35% of caloric intake. (26) Percentage adequacy is represented as individuals who meet or are above the lowest range of AMDR for protein, 10%.
(DOCX)

## Acknowledgments

The authors would like to acknowledge Dr Karen Mumme and the Master of Dietetics students from Massey University, Auckland for their role in matching and compiling nutrient content of all food items from the food diaries to the NZ food composition databases. We would also like to thank Mr Jacob Knight, a Sustainable Nutrition Initiative ™ summer internship student, for his contribution to matching food items to the USDA food composition database.

## Author contributions

**Conceptualization:** Bi Xue Patricia Soh, Nick W. Smith, Pamela R. von Hurst.

**Data curation:** Bi Xue Patricia Soh.

**Formal analysis:** Bi Xue Patricia Soh, Matthieu Vignes, Nick W. Smith.

**Funding acquisition:** Pamela R. von Hurst, Warren C. McNabb.

**Investigation:** Bi Xue Patricia Soh, Pamela R. von Hurst.

**Methodology:** Bi Xue Patricia Soh, Matthieu Vignes, Nick W. Smith.

**Project administration:** Pamela R. von Hurst, Warren C. McNabb.

**Supervision:** Matthieu Vignes, Nick W. Smith, Pamela R. von Hurst, Warren C. McNabb.

**Writing – original draft:** Bi Xue Patricia Soh.

**Writing – review & editing:** Bi Xue Patricia Soh, Matthieu Vignes, Nick W. Smith, Pamela R. von Hurst, Warren C. McNabb.

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
