## [Decision Letter · Decision Letter 0]

7 Feb 2025

Dear Dr. Soh,

Thank you for submitting your manuscript to PLOS ONE. After careful consideration, we feel that it has merit but does not fully meet PLOS ONE’s publication criteria as it currently stands. Therefore, we invite you to submit a revised version of the manuscript that addresses the points raised during the review process.

We look forward to receiving your revised manuscript.

Kind regards,

Ewa Tomaszewska, DVM Ph.D

Academic Editor

PLOS ONE

Journal Requirements:

Lottery Health Project Grant (LHR-2022-185)

PhD stipend from the Riddet Institute, Massey University 

3. Please remove all personal information, ensure that the data shared are in accordance with participant consent, and re-upload a fully anonymized data set. 

Additional Editor Comments :

Dear Authors,

before the acceptance, the text should be improved:

It starts by saying that only 0.74% of the population in the NZ Health Survey identifies as vegan. It goes on to say that “some studies” report a growing trend among young adults and a greater popularity of plant-based diets among women. It’s unclear: is veganism really growing, or is it just plant-based diets in general? Is the New Zealand data representative of the global trend?

“Concerns on nutrient inadequacies accorded by a vegan diet have been highlighted” – awkward wording, better: Concerns about nutrient inadequacies in a vegan diet have been highlighted.

it mentions “long-chain fatty acids such as docosahexaenoic acid” (DHA). However, DHA is not a typical fatty acid in plant foods, and its deficiency is more likely due to a lack of algal sources or supplementation.

The claim that “regular high-quality protein intake throughout the day is essential to ensure IAA quantities are at threshold level” suggests that only high-quality protein can provide adequate amounts of essential amino acids. In fact, if the diet is well-balanced, various plant protein sources can complement each other’s amino acid profiles.

It first states that “few studies evaluated amino acid (AA) intake among vegans,” and later that “diet consisting solely of plant proteins typically provides lower amounts of digestible amino acids.” Since few studies have evaluated amino acid intake among vegans, how can one claim that their intake is inadequate?

“Within a meal context, the presence of at least one limiting IAA halts protein synthesis” is an oversimplification. Metabolic processes do not immediately stop because of one missing amino acid.

"The main aim of this study was to determine protein adequacy among a cohort of vegans in NZ" – but in what aspect? Only quantitative (grams of protein) or also qualitative (amino acid profile, digestibility)?

In the first sentence, veganism is defined as a diet that excludes animal products, but later the term "plant-based diet" (PB diet) is used, which can also include flexitarian or vegetarian diets. It would be necessary to clarify whether PB diet in this text means a strictly vegan diet or a more general plant-based approach.

The text indicates that plant sources of protein have lower protein quality due to the presence of limiting amino acids (IAAs) and anti-nutritional factors. However, there is no explanation as to whether proper planning of a vegan diet actually leads to deficiencies or only increases the risk of their occurrence.

There is also no emphasis on the fact that plant proteins can complement each other, which means that a well-composed vegan diet can provide all the necessary amino acids.

The statement: "Presence of at least one limiting IAA halts protein synthesis and deposition within the body" suggests that protein synthesis stops completely when one amino acid is missing. In fact, the body may be partially adjusting metabolism by recycling amino acids or limiting muscle catabolism.

Food amino acid data were obtained from the USDA database, although the study was conducted in New Zealand. It was not stated whether local products differ significantly in composition, nor whether other databases are available that are adapted to the actual foods consumed in New Zealand.

Normalizing amino acid values from the USDA to the FOODfiles database may lead to errors in estimates if the differences in composition are significant.

The decision to exclude all products providing <0.5 g protein may lead to underestimation of protein and amino acid intake, especially if these foods are consumed frequently.

A protein quality assessment based on the ratio of IAA (mg) to total protein intake is described. However, no mention was made of more standard methods such as PDCAAS (Protein Digestibility Corrected Amino Acid Score) or DIAAS (Digestible Indispensable Amino Acid Score), which are commonly used to assess protein quality.

A definition of normal BMI for the New Zealand population is provided, but it is worth noting that standards may differ for different ethnic groups (e.g. people of Asian descent may have different BMI thresholds for overweight and obesity).

For body fat percentage, thresholds for overweight and obesity are provided, but no reference is made to standards for normal body fat.

There is no mention of possible errors in measuring dietary intake (e.g. underestimation in food diaries), or of the limitations of using a single DXA body composition measurement.

If the mean protein intake meets the EAR for both sexes, but only 56.2% meet the essential amino acid requirements, this suggests a serious protein quality problem. However, later on, it is stated that “meeting protein quality in a vegan diet can be likely achieved when total protein requirement is met” (lines 505–507), which contradicts the previous conclusion.

If the group had a higher body mass and at the same time consumed less energy, then either there is a serious error in the measurements (e.g. underreporting of energy intake) or these individuals must have had a lower metabolic rate, which is not explained.

It is suggested here that people with a normal BMI meet the amino acid requirements more easily, but a few lines later (lines 435–438) it is stated that people with a higher BMI may have a lower protein requirement. There is no clear explanation of exactly how body composition affects protein requirements.

lines 505–507 If so, why was it previously found that 43.8% of individuals did not meet the standards for IAAs despite having adequate protein intake? This suggests that protein quantity alone does not guarantee protein quality, so this paradox needs to be better explained.

with best regards

Ewa Tomaszewska

Reviewers' comments:

Reviewer's Responses to Questions

**Comments to the Author**

1. Is the manuscript technically sound, and do the data support the conclusions?

Reviewer #1: Yes

Reviewer #2: Yes

2. Has the statistical analysis been performed appropriately and rigorously?

Reviewer #1: Yes

Reviewer #2: Yes

3. Have the authors made all data underlying the findings in their manuscript fully available?

Reviewer #1: Yes

Reviewer #2: Yes

4. Is the manuscript presented in an intelligible fashion and written in standard English?

Reviewer #1: Yes

Reviewer #2: Yes

Reviewer #1: The manuscript was well written. The primary and secondary objective were answered with relevant and sound method and analysis. Discussion had noted the main findings and been elaborated accordingly..

Reviewer #2: The manuscript entitled “Evaluation of protein intake and protein quality in New Zealand vegans” presents a proposal of relevance to those who choose a vegan lifestyle. Although it is a small percentage worldwide, it is important to generate knowledge that allows to know that their diet has an optimal quality.

The main observation made by this reviewer when reading this manuscript was the processing and collection of data. The dietary record may lack reliability, since the proportions may be underestimated, or not report all the foods consumed due to forgetfulness or fatigue. On the other hand, not having indicators specific to the sample under study and reusing those used in other populations can be a disadvantage.

Although the aforementioned may be a disadvantage of the data presented in this study, the authors take it up in the limitations section and provide valid arguments for the conclusions.

It is suggested that the authors revise the wording of “Data processing” as it is confusing.

**Do you want your identity to be public for this peer review?** For information about this choice, including consent withdrawal, please see our Privacy Policy

Reviewer #1: No

Reviewer #2: No

---

## [Author Response · Author response to Decision Letter 0]

23 Feb 2025

Response to reviewers have been organised and attached in a separate document that has been included in the attached files

---

## [Editor Report · Decision Letter 1]

26 Feb 2025

Evaluation of protein intake and protein quality in New Zealand vegans

PONE-D-24-52340R1

Dear Dr. Bi Xue Patricia Soh,

We’re pleased to inform you that your manuscript has been judged scientifically suitable for publication and will be formally accepted for publication once it meets all outstanding technical requirements.

Kind regards,

Ewa Tomaszewska, DVM Ph.D

Academic Editor

PLOS ONE
---

## [Editor Report · Acceptance letter]

PONE-D-24-52340R1

PLOS ONE

Dear Dr. Soh,

I'm pleased to inform you that your manuscript has been deemed suitable for publication in PLOS ONE. Congratulations! Your manuscript is now being handed over to our production team.

Kind regards,

on behalf of

Professor Ewa Tomaszewska

Academic Editor

PLOS ONE